# AdvTG: An Adversarial Traffic Generation Framework to Deceive DL-Based Malicious Traffic Detection Models

## Abstract

Deep learning-based (DL-based) malicious traffic detection methods are effective but vulnerable to adversarial attacks. Existing adversarial attack methods have shown promising results when targeting traffic detection models based on statistics and sequence features. However, these methods are less effective against models that rely on payload analysis. The main reason is the difficulty in generating semantic, compliant, and functional payloads, which limits their practical application.

In this paper, we propose AdvTG, an adversarial traffic generation framework based on the large language model (LLM) and reinforcement learning (RL). Specifically, AdvTG is designed to attack various DL-based detection models across diverse features and architectures, thereby enhancing the generalization capabilities of the generated adversarial traffic. Moreover, we design a specialized prompt for payload generation tasks, where functional fields and target types are supplied as input, while non-functional fields are generated to produce the mutated traffic. This fine-tuning endows the LLM with task comprehension and traffic pattern reasoning abilities, allowing it to generate traffic that remains compliant and functional. Furthermore, leveraging RL, AdvTG automatically selects traffic fields that exhibit more robust adversarial properties. Experimental results show that AdvTG achieves over 40% attack success rate (ASR) across six detection models on four base datasets and two extended datasets, significantly outperforming other adversarial attack methods.

## CCS Concepts

• **Security and privacy → Artificial immune systems**.

## Keywords

Malicious Traffic Detection, Adversarial Attacks, Large Language Model, Reinforcement Learning

## 1 INTRODUCTION

Malicious traffic detection is an essential approach for detecting malicious activities in networks. The traditional method extracts the key fields of malicious traffic as signature rules [13, 18, 41]. However, as attack tactics continue to evolve, signature-based detection methods are increasingly less effective at detecting unknown or mutated malicious traffic. With the development of artificial intelligence, deep learning (DL) has seen widespread adoption in cybersecurity. Most methods train DL-based models with large amounts of traffic to detect traffic automatically [20, 57, 62]. At present, DL-based malicious traffic detection has become a familiar and effective technique in both academic and industrial settings [2, 32, 42].

Unfortunately, DL-based models are vulnerable to adversarial attacks [12, 34, 46, 59, 63]. In the field of network security, adversarial samples have even more severe consequences [2]. Traffic features can be modified to specifically target detection models, causing the models to produce incorrect classification results. Recently, numerous adversarial methods have been developed to exploit vulnerabilities in DL-based traffic detection systems and reveal their weaknesses.

Existing adversarial attacks in the traffic domain can be categorized into three types. **Statistical feature attacks** [17, 33, 49] modify statistical features (e.g., flow duration and average packet size) but cannot be easily mapped back to actual traffic. **Sequence feature attacks** [22, 30, 39, 48, 50] alter temporal and spatial sequence features (e.g., packet intervals, packet lengths) but payload-based detection models remain effective. **Content attacks** [10, 35, 51, 58] directly modify traffic packets to evade detection. The first two types of attacks have already achieved considerable success in adversarial traffic attacks; however, they fail to deceive models which detect based on payload. Compared to these attacks, content attacks pose more significant challenges, but they are also more disruptive. By modifying the payload, these attacks can deceive payload-based detection models [53, 55], allowing malicious activities to go unnoticed. In this study, we focus solely on content attacks, specifically targeting DL-based detection models that rely on payload analysis, and exclude the first two types of attacks from consideration.

Existing adversarial attacks on DL-based traffic detection are limited to specific scenarios and rely on impractical assumptions. Table 1 provides a comparative overview of these adversarial attack methods within the traffic detection domain. The key challenges can be summarized into three categories.

- **Generality.** Traffic features should be mapped to real-world traffic spaces rather than generating non-existent features. Adversarial attacks should demonstrate generalization, being able to succeed across various extracted features and model architectures.
- **Availability.** The generated traffic must adhere to strict protocol compliance while maintaining its functionality. Functional fields are those that play a critical role in the execution and routing of traffic packets. These fields directly influence how the server processes the request or how the client interacts with the server. For example, the request line and the payload that executes malicious commands in HTTP traffic are considered functional fields.
- **Payload Generation.** This challenge is specific to content attacks. The generated adversarial traffic should consist of complete packets, and remain semantic traffic.

Benefiting from advancements in large language models (LLMs), which significantly enhance both semantic understanding and text generation capabilities. Therefore, we are able to focus on traffic content attacks aimed at generating mutated traffic, which can deceive existing content-based detection models.

In this paper, we propose AdvTG, an adversarial traffic generation framework designed to deceive DL-based detection models. The framework consists of three key stages.

**Table 1: The comparison with the existing methods of adversarial attacks in traffic.**

| Attack Types | Target Models | Attack Methods | Generality | | | Availability | | Payload-generation | |
|---|---|---|---|---|---|---|---|---|---|
| | | | Mappable Data | Black-Box Models | Various Features | Protocol Compliance | Remaining Functionality | Complete Packet | Semantic Content |
| Statistical Feature Attacks | ML & DL | IDSGAN [33] | × | × | × | × | × | - | - |
| | | DIGFuPas [17] | × | × | × | × | ✓ | - | - |
| | Multi-Source | Bars[49] | × | ✓ | × | × | ✓ | - | - |
| | | Multiple Methods [28] | × | ✓ | × | × | × | - | - |
| | | Flow Statistics [37] | ✓ | × | × | ✓ | × | - | - |
| Sequence Feature Attacks | ML & DL | Prism [30] | ✓ | × | ✓ | ✓ | ✓ | - | - |
| | | RL [48] | ✓ | ✓ | ✓ | ✓ | ✓ | - | - |
| | | ProGen [50] | ✓ | × | ✓ | ✓ | ✓ | - | - |
| | Multi-Source | Blanket [39] | ✓ | × | × | × | ✓ | - | - |
| | | GA+GAN [22] | ✓ | ✓ | × | × | ✓ | - | - |
| Content Attacks | ML & DL | Attack-GAN [10] | × | ✓ | × | × | × | ✓ | × |
| | | GA [58] | ✓ | ✓ | × | × | ✓ | × | × |
| | Multi-Source | Text Attack [35] | ✓ | ✓ | × | × | ✓ | × | × |
| | | Fuzzy Attack [51] | ✓ | ✓ | × | ✓ | ✓ | × | ✓ |
| | | **AdvTG** | ✓ | ✓ | ✓ | ✓ | ✓ | ✓ | ✓ |

First, we train multiple DL-based models using both image-based and text-based features extracted from payload. These models serve as targets for our adversarial attacks. Simultaneously, these models serve as reward models during the adversarial generation process, providing feedback on how effectively the generated traffic deceives detection. Using diverse features and model architectures improves AdvTG's ability to generate adversarial traffic with improved generalization.

Next, we fine-tune the LLM with a traffic-specific prompt format. Instructions define the traffic categories, and functional fields are provided as input while the LLM generates non-functional fields to create a complete payload. The fine-tuning approach ensures the generated traffic is compliant, functional, and semantically rich.

Finally, reinforcement learning (RL) is applied to optimize the adversarial traffic generation process. The detection models provide feedback on whether each generated traffic sample effectively deceives detection, enabling the LLM to refine and enhance its outputs based on feedback continually. This adaptive process enables the LLM to optimize adversarial non-functional fields, making the generated traffic harder for detection models to detect.

- We propose AdvTG[1], an adversarial traffic generation framework based on the LLM and RL to deceive DL-based malicious traffic models.
- We introduce a tailored fine-tuning process for the LLM, designed to enhance its understanding for traffic generation tasks, enabling it to generate semantic traffic. By using prompts that specify the expected traffic type and functional fields, the LLM generates traffic by altering non-functional fields while preserving both functionality and protocol compliance.
- AdvTG leverages RL to optimize adversarial traffic generation by continuously generating mutated traffic based on

feedback from detection models. This adaptive process enables the LLM to iteratively improve its outputs iteratively, identifying the optimal adversarial non-functional fields and making the generated traffic increasingly difficult for detection models to identify.
- Experimental results show that AdvTG achieves over 40% attack success rate (ASR) across six detection models on four base datasets and two extended datasets, significantly outperforming other adversarial attack methods.

## 2 BACKGROUND AND MOTIVATION

### 2.1 Scenarios

We clarify our scenarios by answering the following key questions.

**A. Why are we targeting content adversarial attacks?**

Existing adversarial traffic attack methods work by altering the direction of packets or employing techniques such as truncation and padding. These approaches have demonstrated vulnerabilities in many feature-based traffic detection models. However, such changes are easily detected by models that analyze deeper features, such as the payload. Our goal is to generate adversarial mutated traffic, aiming to evaluate the robustness of payload-based detection models. Traffic data is structured and frequently contains redundant information; however, upper-layer protocols typically carry more meaningful data compared to lower-layer protocols like IP and TCP. Therefore, we focus content attacks on application-layer protocols.

**B. Why are we focusing on HTTP protocol?**

HTTP/HTTPS protocols have long been primary communication protocols on the internet and are key vectors for malicious activities. Many attacks, such as web exploitation, vulnerability exploitation, and C&C communications, are carried out via these protocols. We focus on adversarial attacks on plaintext traffic, excluding encrypted traffic. The main reason is that encrypted data is transformed into unpredictable ciphertext, and any modification

---
[1]https://github.com/TrafficDetection-art/AdvTG

can corrupt the data, leading to incorrect plaintext upon decryption. However, plaintext traffic detection research can extend to encrypted traffic, as defenders can use decryption techniques for effective monitoring. Essential security tools like Endpoint Detection and Response (EDR), Web Application Firewalls (WAF), and Internet Information Services (IIS) rely on plaintext analysis to enhance security.

### C. Where to add perturbations to achieve attacks?

HTTP traffic has a strict header specification with many fields, as shown in Fig. 1. Both benign and malicious traffic exhibit certain regularities in the header fields they utilize, with noticeable differences between the two. These fields are commonly used as key features in malicious traffic detection. We can categorize the packet into functional and non-functional fields based on their necessity, with their differences detailed in an example in § A. In brief, functional fields cannot be altered, whereas non-functional fields can be modified. By targeting the non-functional fields, we can attack the detection model without compromising the compliance and functionality of the HTTP traffic.

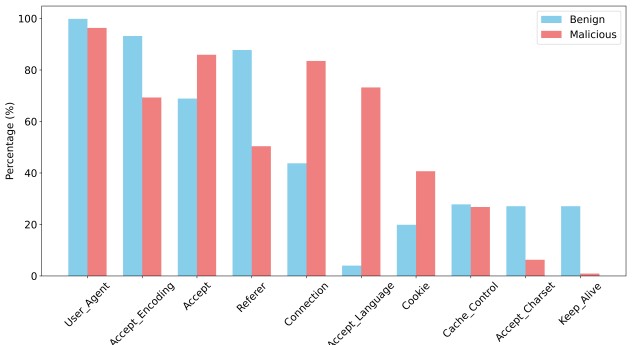

**Figure 1: Comparison of header fields between benign traffic and malicious traffic.**

## 2.2 THREAD MODEL

We define threat model based on previous work [5].

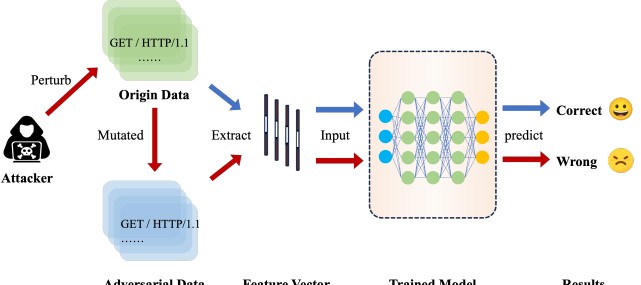

**Figure 2: The scenarios of adversarial attacks.**

**Attacker's Goal.** The attacker seeks to generate adversarial traffic to deceive DL-based detection models, leading to misclassification, as shown in Fig. 2. The attacker produces complete payloads while ensuring they remain protocol-compliant and functional.

**Attacker's Knowledge.** The attacker has no prior knowledge of the detection model architecture or the extracted features. Instead, the attacker inputs the generated traffic into a black-box detection, which extracts features and outputs classification results. These results serve as a reward signal, enabling the attacker to generate more challenging adversarial traffic based on RL automatically.

**Attacker's Capability.** The attacker can only modify the test dataset and cannot alter the training dataset, thus preventing data poisoning attacks [6]. The attacker leverages the advanced LLM to generate adversarial traffic that remains both semantic and functional.

## 3 OVERVIEW

In this section, we propose AdvTG, targeting DL-based malicious traffic detection models. Fig. 3 shows three phases of AdvTG. (i) We train multiple DL-based malicious traffic detections with different traffic features and model architectures using supervised learning. (ii) The domain-specific LLM is fine-tuned using large amounts of HTTP traffic based on self-supervised learning. (iii) The domain-specific LLM is further fine-tuned to carry out adversarial attacks using RL based on the feedback from the malicious traffic detection in the first stage.

**Detection Model Training.** We train multiple malicious traffic detection models as adversarial attack targets. These models are trained on small datasets and demonstrate a certain level of generalization in detecting malicious HTTP traffic. The detection models utilize both image-based and text-based features, incorporating a range of standard DL-based architectures alongside prominent models from the academic field of malicious traffic detection [14, 16, 29, 36, 53, 64].

The trained models are used as reward models for adversarial generation through RL. Specifically, these classification models evaluate the quality of the traffic generated by the LLM. Traffic that successfully deceives the detection model receives a higher score, while easily detected traffic is given a lower score.

**Domain-Specific LLMs Fine-tuning.** We fine-tune the general-purpose LLM using a large set of domain-specific data. This process helps the model learn HTTP traffic patterns and formats, improving its ability to generate and understand data within this domain while also enhancing its distinction between benign and malicious traffic.

We propose a traffic-specific prompt format where instructions define the traffic categories. Functional fields are given as input, while non-functional fields are generated to form a complete traffic packet.

**RL-based Adversarial Generation.** Reinforcement learning is employed to optimize the adversarial attack generation strategy. The reward model, trained in the first phase, evaluates the outputs of the LLM, and these scores are used to update the LLM's parameters.

In practice, a batch of prompts is randomly sampled, and the fine-tuned LLM generates traffic packets. The generated traffic is then input into the detection model to receive feedback, where the reward model assigns a reward representing the overall quality of the generated payload. Once the final reward for the word sequence is obtained, it is propagated backwards through the sequence, treating each word as a time step. The objective is to train the LLM to

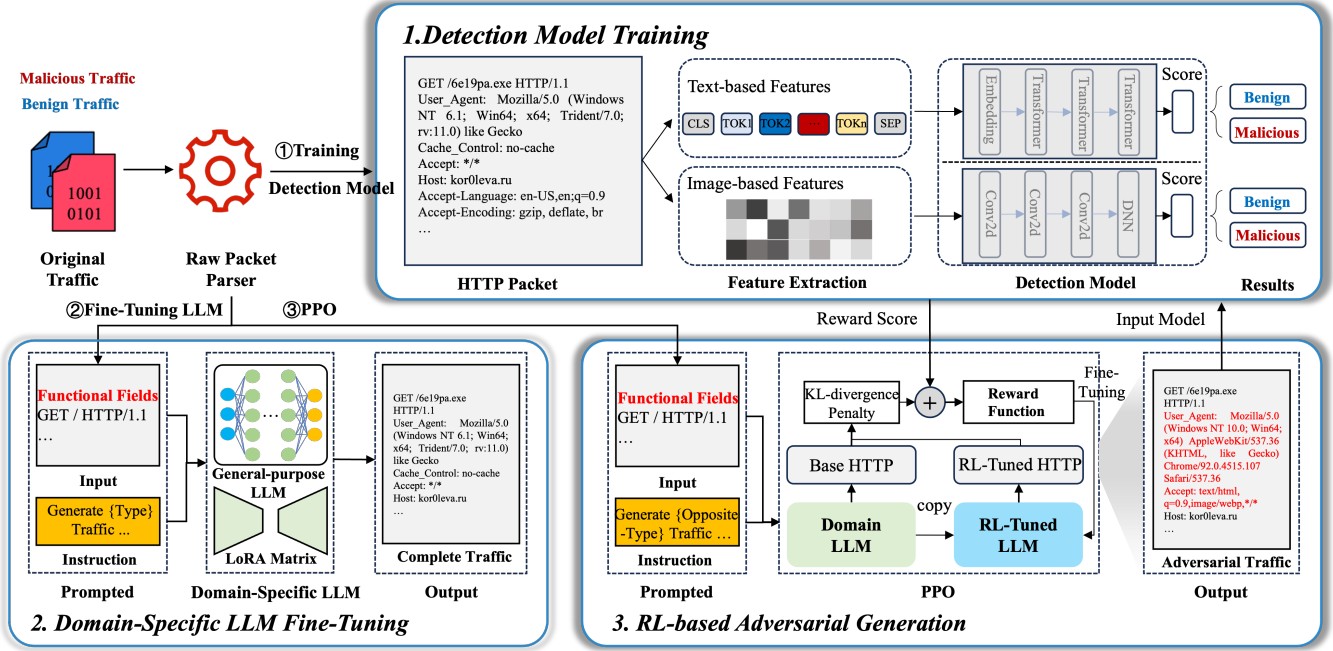

**Figure 3: The overview of AdvTG.**

produce high-reward outputs that align with the reward model and represent high-quality responses.

## 4 DETECTION MODEL TRAINING

According to the threat model, we train multiple detection models with probabilistic outputs through supervised training. We design multiple feature extractors and models to verify the generalization of adversarial attacks to different feature models. The feature extractor converts the traffic into a continuous or discrete numerical sequence that can be input into the model. At present, in the field of deep learning, detection models for traffic payload mainly include two types. One is transforming traffic into images, where the byte stream data of the traffic is converted into a two-dimensional grayscale image. This representation is then classified using deep learning models [36, 53]. The other approach treats traffic as textual data, converting traffic into vector representations using word embedding techniques [26, 52].

### 4.1 Image-based Features

Converting network traffic into images enables the capture of spatial and local patterns within the data. By representing each byte as a pixel, this method leverages image analysis techniques, such as CNN, to enhance detection accuracy. Eq. 1 and Eq. 2 outline the process of transforming network traffic $T$ into an image. The traffic is first divided into individual bytes and normalized to a continuous sequence $T'$ between 0 and 1. This sequence is then reshaped into a 2D image of size $m \times n$. If the traffic exceeds the predefined image size, the excess data is truncated; otherwise, padding with zeros is applied to match the required dimensions.

$$T' = \frac{T}{255}, \quad T = [b_1, b_2, \ldots, b_n], \; b_i \in [0, 255] \quad (1)$$

$$I(i, j) = T'_{(i-1) \times n + j} \; (i \in [1, m], j \in [1, n]) \quad (2)$$

The extracted features are input into a DL-based model, which outputs classification probability scores. In this work, we utilize various DL-based model architectures, including the classic CNN and other commonly used models in the field of image processing.

### 4.2 Text-based Features

Transforming traffic into text highlights its semantic payload and sequential patterns, aiding detection through semantic analysis. In Eq. 3, $T$ denotes the traffic data, which is tokenized into $(w_1, w_2, \ldots, w_n)$. Each token $tokenizer(T)_k$ is mapped to an index $t_k$ via a vocabulary function $V$, assigning a unique identifier to preprocess the data for input into a DL-based model.

$$t_k = V(tokenizer(T)_k), \quad k = 1, 2, \ldots, n \quad (3)$$

Once the token sequence is processed by the tokenizer, it passes through the complex DL-based model, which outputs classification probabilities. In this paper, we employ various model architectures, including the classic LSTM and BERT [14].

### 4.3 Target Model

The feature models mentioned above serve as target models for our adversarial attacks. This process follows a black-box attack scenario, where the attacker has no knowledge of the target model's features or architecture and relies solely on the output scores. In the third phase, RL is employed to fine-tune the LLM, using these models

as reward models. The reward model guides the LLM to generate payloads that more effectively challenge and deceive the detection model.

## 5 DOMAIN-SPECIFIC LLM FINE-TUNING

In this section, we fine-tune a general-purpose LLM using traffic datasets to improve its performance in traffic generation tasks. While advancements in LLMs have significantly enhanced text generation across general language understanding tasks [8, 43, 44], these LLMs often struggle in domain-specific applications due to a lack of specialized knowledge and contextual understanding. Fine-tuning on specialized datasets effectively addresses this limitation. The LLM is able to better understand and generate text that fits the specific domain tasks.

We employ the Parameter-Efficient Fine-Tuning (PEFT) [31] to improve efficiency. Specifically, we utilize Low-Rank Adaptation (LoRA) [25], which minimizes the number of trainable parameters by focusing on low-rank matrix adaptation during fine-tuning. As shown in Eq. 4 and Eq. 5, $h$ represents the fine-tuned model parameters, $W_0$ denotes the original pre-trained parameters, and matrices $A$ and $B$ serve as the low-rank down-projection and up-projection matrices, respectively, which are updated during fine-tuning. While full-parameter fine-tuning requires updating a large number of parameters, LoRA learns the low-rank matrix $BA$ while keeping the pre-trained weights $W_0$ frozen, which significantly reduces the trainable parameter counts, and reducing GPU consumption.

As shown in Fig. 3, we design prompts specifically for traffic generation. The instruction set guides the generation of specific traffic categories, with the input consisting of the functional fields. The LLM generates the non-functional fields, forming a complete traffic packet. The prompts are specifically designed to keep the functional fields unchanged while modifying the non-functional fields to create mutated traffic. Having learned the structural patterns of both benign and malicious traffic during fine-tuning, the LLM is able to generate corresponding traffic based on instructions.

$$B \in \mathbb{R}^{d \times r}, \quad A \in \mathbb{R}^{r \times k} \tag{4}$$

$$h = W_0 x + \Delta W x = W_0 x + BA x \tag{5}$$

## 6 RL-BASED ADVERSARIAL GENERATION

In this section, we employ Proximal Policy Optimization (PPO) [45] to fine-tune a LLM [40]. We propose a novel approach that builds upon the previously fine-tuned LLM in § 5, applying PPO for further fine-tuning. Specifically, the LLM, already adapted to the generation task, is now tasked with generating adversarial traffic to deceive detection models. This process treats the detection models as black-box systems that only output scores, with the LLM generating payloads and receiving reward scores based on the detection models' evaluations (§ 4).

As shown in Fig. 3, the prompt set $x$ consists of two parts: an instruction and functional fields. The functional fields are provided as input, while the instruction directs the LLM to generate traffic from the opposite class. For example, when the input consists of functional fields for malicious traffic, the instruction requires the output to be benign traffic. The LLM generates benign traffic $y$ that deviate from the expected behavior based on the prompt $x$. This

allows the attacker to craft complete adversarial traffic tailored to each input's functional fields.

We design a novel reward mechanism that makes PPO suitable for adversarial attack tasks. The PPO process involves two LLMs: the domain-specific LLM, with frozen parameters, and the RL-Tuned LLM, which is updated during training. The purpose of this setup is to refine the RL-Tuned LLM's ability to generate adversarial traffic while preserving the foundational behavior learned by the Domain LLM. The RL-Tuned LLM's outputs are evaluated using multiple reward models trained in the first phase. As shown in Eq. 6, each reward model provides a reward score, $r_\theta(x, y)$, which can be interpreted as a variant of the cross-entropy function, referred to as negative cross-entropy. Traditional cross-entropy minimizes the difference between the predicted value $D_i(y)$ and the label $L_x$. In contrast, negative cross-entropy increases the reward as the prediction deviates further from $L_x$, encouraging the generation of adversarial examples. The final reward score is calculated as the average of multiple reward model $D_i(y)$ outputs. This aggregation provides a more balanced and comprehensive evaluation, ensuring that the adversarial traffic generated by the RL-Tuned LLM effectively challenges a range of detection models.

In addition to the averaged reward score, the PPO includes a KL-divergence penalty. This penalty ensures that while the RL-Tuned LLM is optimized for adversarial performance, its outputs do not deviate excessively from the behavior of the domain-specific LLM. The KL-divergence term helps maintain consistency in the generation process, preventing the RL-Tuned LLM from generating traffic that is too different from the original model's behavior. As shown in Eq. 7, the PPO process strikes a balance between generating highly adversarial traffic and retaining the underlying structure and coherence of the original model by combining multiple reward models with the KL-divergence term.

$$r_\theta(x, y) = -\frac{1}{N} \sum_{i=1}^{N} \left[ (1 - L_x) \log(D_i(y)) + L_x \log(1 - D_i(y)) \right] \tag{6}$$

$$R(x, y) = r_\theta(x, y) - \beta \log \left[ \pi_\phi^{\text{RL}}(y \mid x) / \pi^{\text{SFT}}(y \mid x) \right] \tag{7}$$

In general, we leverage a novel reward mechanism and balance it with a KL-divergence penalty, enabling the RL-Tuned LLM to generate mutated traffic that detection models struggle to detect.

## 7 EXPERIMENTAL EVALUATION

### 7.1 Experiment Setup

**Testbed.** We deploy AdvTG in Ubuntu 22.04.1 with Python3.11 based on a testbed built upon Supermicro servers with one AMD EPYC 7532 CPU, one NVIDIA 4090, 256GB MEM.

**Datasets.** We mainly use a variety of open-source malicious traffic datasets and self-collected datasets. The details of the datasets are introduced in § C.

**Targeted Detection Models.** We construct three different detection models, each applied separately to both image-based and text-based feature extraction methods.

Image-based Detection Models.

- **MalTraffic** [53] first maps traffic to grayscale images and trains it based on the CNN model.

- **RBRN** [64] learns the deep features of traffic through coding and relational networks and detects malicious traffic by calculating the similarity of traffic pairs.
- **DeepMal** [36] leverages a combination of CNN and LSTM, enabling better detection of malware traffic.

Text-based Detection Models.

- **TextCNN** [29] applies CNN to text data. It works by sliding convolution filters over word embeddings to detect text.
- **DeepLog** [16] is commonly used as an anomaly detection model in the cybersecurity field, and it uses LSTM-based deep learning to detect anomalous logs.
- **BERT** [14] is a transformer-based model designed specifically for NLP tasks and is advantageous for long text classification tasks.

**LLM Architecture.** We select Meta-Llama-3-8B as the base LLM for fine-tuning and adversarial sample generation [1].

**Attack Baselines.** We select text-based adversarial attack models and do not consider image domain, including some methods mentioned in § 1. This is because these attacks introduce byte-level perturbations that either violate network protocol specifications or fail to generate complete traffic. Therefore, we compare three text-based adversarial attack methods.

- **TextFooler (T-F)** [27] generates adversarial attacks by replacing key words with semantically similar alternatives to change model predictions.
- **WordLevel (W-L)** [61] optimizes word substitutions to mislead models, preserving the text's meaning and fluency.
- **BAE** [21] uses BERT to generate adversarial examples through word substitution and insertion, maintaining the text's original semantics.

In addition to the text adversarial attack methods, we employ the **Random Substitution (R-S)** method. Specifically, we randomly select unrelated fields from different categories to replace the original non-functional fields. For instance, non-functional fields in malicious traffic are substituted with those from benign traffic, thereby deceiving the detection model.

**Metrics.** We evaluate our approach primarily from both detection effectiveness and attack effectiveness perspectives.

- **Detection Effectiveness.** We mainly use precision (P), recall (R), and F1 score which they are most widely used in the literature [7, 19, 38].
- **Attack Effectiveness.** We use the attack success rate (ASR) as the primary metric to evaluate the effectiveness of attacks [4, 9, 54], measuring the proportion of successful attacks in deceiving the detection model as shown in Eq. 8.

$$ASR = \frac{N_{\text{success}}}{N_{\text{total}}} \quad (8)$$

## 7.2 Evaluation

The following sections present a comprehensive evaluation of detection models and adversarial attack methods across several datasets. First, we evaluate the effectiveness of six detection models using both text and image features, and evaluate attack performance based on ASR. Then, we evaluate the adversarial traffic compliance

before and after fine-tuning using a compliance evaluation algorithm. In addition, ablation studies are conducted to analyze the impact of RL on improving ASR. Finally, extended datasets are used to verify the generalization capabilities of AdvTG.

Table 2 summarizes the performance of detection models and attack methods. The left three columns (P, R, F1) evaluate the detection accuracy of six models across two feature types, while the right five columns present the attack success rate (ASR) to evaluate the effectiveness of various attack methods.

**Detection Effectiveness.** We evaluate the effect of detection models. We train models based on text and image features based on training sets and test their detection effects on test sets. As shown in Table 2, the F1 score on the test set is higher than 0.96 regardless of the characteristics of the model used. Noteworthy, the text-based models all reach above 0.98, which is better than the image-based models. This shows that the text feature is more effective at detecting the plaintext payload and can better identify the relevant patterns.

**Attack Effectiveness on the Base Datasets.** We evaluate the impact of adversarial traffic on detection models by measuring the ASR of four traffic. As shown in Table 2, AdvTG consistently achieve over 40% ASR across four datasets, whereas text-based attacks result in less than 10% ASR. This is because traditional methods focus on altering a few keywords, which is insufficient to evade detection. Our approach, leveraging LLM, modifies multiple fields while ensuring the traffic remains both semantic and functional, producing a diverse range of mutated traffic templates that deceive detection models.

Additionally, since text-based adversarial attacks are limited to text domains, they cannot target models relying on image features. In contrast, AdvTG provides general adversarial attacks, effective across models with any content-based features. Furthermore, the R-T attack also shows a low ASR, below 22.27%, indicating that replacing non-functional fields alone is not enough to mislead the models. This underscores the generalization capability of detection models. By using RL with multiple reward feedback loops, our approach identifies fields most likely to deceive the model, resulting in a higher ASR.

**Traffic Compliance.** We develop an algorithm, detailed in § D, to evaluate the compliance of generated traffic. We compare the compliance proportion of traffic generated by the general-purpose LLM with that of the fine-tuned LLM. As shown in Table 3, traffic generated before fine-tuning has a compliance proportion of around 20% across the four datasets. After fine-tuning, this proportion increases significantly to over 80%, demonstrating that the methods proposed in § 5 effectively improve traffic compliance for real-world applications.

**Ablation Experiments.** As shown in Fig. 4, we conduct adversarial generation experiments on a general-purpose LLM without fine-tuning, a fine-tuned domain-specific LLM, and an RL-Tuned LLM. The domain-specific LLM shows weak adversarial capabilities. However, after applying RL, the ASR significantly improves, demonstrating the effectiveness of RL. Additionally, while some general-purpose LLMs without fine-tuning achieve relatively high

**Table 2: The adversarial effectiveness of DL-based traffic detection on the base datasets.**

(a) *CICIDS2017*

| Feature Extractor | ML Classifier | Detection | | | Attack (ASR)—*higher is better* | | | | |
|---|---|---|---|---|---|---|---|---|---|
| | | P | R | F1 | T-F | W-L | BAE | R-T | **AdvTG** |
| **Text** | TextCNN | 0.99 | 1 | 0.99 | 0.07% | 0.13% | 0.60% | 0.84% | **69.53%** |
| | DeepLog | 0.99 | 0.99 | 0.99 | 2.01% | 2.02% | 2.61% | 5.25% | **66.41%** |
| | BERT | 0.99 | 0.99 | 0.99 | 0.24% | 0.08% | 2.42% | 3.94% | **67.18%** |
| **Image** | MalTraffic | 0.98 | 0.99 | 0.98 | - | - | - | 20.13% | **42.78%** |
| | RBRN | 0.98 | 0.99 | 0.99 | - | - | - | 20.08% | **42.17%** |
| | DeepMal | 0.98 | 0.99 | 0.98 | - | - | - | 16.68% | **43.69%** |

(b) *CICIoT2023*

| Feature Extractor | ML Classifier | Detection | | | Attack (ASR)—*higher is better* | | | | |
|---|---|---|---|---|---|---|---|---|---|
| | | P | R | F1 | T-F | W-L | BAE | R-T | **AdvTG** |
| **Text** | TextCNN | 0.99 | 0.99 | 0.99 | 0.13% | 0.74% | 0.60% | 1.08% | **69.53%** |
| | DeepLog | 0.98 | 0.99 | 0.99 | 2.41% | 2.43% | 3.02% | 7.44% | **66.41%** |
| | BERT | 0.99 | 0.99 | 0.99 | 0% | 0.35% | 2.69% | 5.33% | **67.19%** |
| **Image** | MalTraffic | 0.96 | 0.99 | 0.97 | - | - | - | 20.87% | **66.99%** |
| | RBRN | 0.96 | 0.99 | 0.98 | - | - | - | 20.68% | **64.08%** |
| | DeepMal | 0.96 | 0.99 | 0.98 | - | - | - | 20.87% | **64.56%** |

(c) *Malware*

| Feature Extractor | ML Classifier | Detection | | | Attack (ASR)—*higher is better* | | | | |
|---|---|---|---|---|---|---|---|---|---|
| | | P | R | F1 | T-F | W-L | BAE | R-T | **AdvTG** |
| **Text** | TextCNN | 0.99 | 0.99 | 0.99 | 0.20% | 0.68% | 0.40% | 1.18% | **65.62%** |
| | DeepLog | 0.98 | 0.99 | 0.99 | 5.35% | 5.35% | 6.38% | 7.26% | **59.37%** |
| | BERT | 0.99 | 0.99 | 0.99 | 0.41% | 0.36% | 2.80% | 5.53% | **61.71%** |
| **Image** | MalTraffic | 0.95 | 0.99 | 0.97 | - | - | - | 20.87% | **67.47%** |
| | RBRN | 0.96 | 0.99 | 0.98 | - | - | - | 22.27% | **67.96%** |
| | DeepMal | 0.96 | 0.98 | 0.97 | - | - | - | 19.32% | **66.50%** |

(d) *APT*

| Feature Extractor | ML Classifier | Detection | | | Attack (ASR)—*higher is better* | | | | |
|---|---|---|---|---|---|---|---|---|---|
| | | P | R | F1 | T-F | W-L | BAE | R-T | **AdvTG** |
| **Text** | TextCNN | 0.99 | 0.99 | 0.99 | 0.11% | 0.79% | 0.14% | 0.76% | **48.43%** |
| | DeepLog | 0.99 | 0.99 | 0.99 | 2.64% | 2.98% | 2.64% | 5.28% | **41.41%** |
| | BERT | 0.99 | 0.99 | 0.99 | 0.10% | 0% | 1.60% | 3.87% | **50.29%** |
| **Image** | MalTraffic | 0.98 | 0.99 | 0.98 | - | - | - | 14.20% | **47.57%** |
| | RBRN | 0.98 | 0.99 | 0.99 | - | - | - | 15.42% | **49.18%** |
| | DeepMal | 0.97 | 0.99 | 0.98 | - | - | - | 15.09% | **49.51%** |

**Table 3: Proportion of Compliance in Generated Traffic Before and After Fine-tuning**

| Datasets | Before Fine-tuning | After Fine-tuning |
|---|---|---|
| CICIDS2017 | 29.68% | 89.16% |
| CICIoT2023 | 16.40% | 82.75% |
| Malware | 39.06% | 87.54% |
| APT | 28.90% | 88.67% |

ASR, their generated traffic often fails to meet compliance, as previously mentioned.

**Attack Effectiveness on the Extended Datasets.** We evaluate the generalization capabilities of AdvTG on two extended datasets, CICAPT-IoT2024 and APT2024. The two datasets are used solely for testing and not involved in training or LLM fine-tuning. As shown in Table 4, the detection models demonstrate strong performance, with F1 consistently above 0.92 across both text-based and image-based detection. After applying adversarial attacks, our method, AdvTG, consistently achieves higher ASR compared to other attacks, with over 40% ASR on CICAPT-IoT2024 and over 50% on APT2024. This indicates that AdvTG generalizes well across different datasets, maintaining a high level of adversarial effectiveness.

## 8 DISCUSSION

In this section, we discuss some potential limitations and challenges of AdvTG.

**Adversarial Attacks against LLM.** LLMs have gradually been applied to cybersecurity, with their unique language understanding capabilities offering an advantage in analyzing payload. As shown in Table 5, we conducted experiments using general-purpose models, ChatGPT 3.5 and ChatGPTs, on APT and malware datasets. Due to the lack of fine-tuning, the F1-scores of both detection models were around 0.6. After generating adversarial samples with AdvTG, the ASR reached only 15%. LLMs possess stronger robustness than DL-based models, because of their better interpretability. In the future, it will be necessary to train detection models specifically based on LLMs rather than relying on general-purpose models and to further verify the robustness of LLMs.

**Encrypted Traffic Detection.** Encrypted traffic detection is an important direction [11, 15]. Many encrypted detection models [24, 32, 56] combine the analysis of sequence features, such as packet size, with an encrypted payload. Therefore, further investigation is needed to determine whether adversarial traffic generated by AdvTG retains its evasive properties after encryption. Exploring the potential to deceive encrypted traffic detection models by manipulating both sequence features and payload presents a valuable avenue for future exploration.

## 9 RELATED WORK

There are several related works in the fields of malicious traffic detection and adversarial attacks.

**Malicious Traffic Detection.** Malicious traffic detection based on deep learning has become a widely used technology. Based on content features, HSTF-Model extracts information from payload [55], and HMCD-Model adds GAN-based enhancement technology for more scenarios [60]. Based on sequence features, ContraMTD uses contrastive learning to learn the relationship between local/global interaction features [23] and Trident transforms known/new class recognition problems into multiple independent single-class learning tasks in traffic detection [63].

**Adversarial Attacks on Traffic Detection.** As shown in Table 1, these methods mainly involve three types of technologies: statistical feature attacks, sequence feature attacks, and content attacks. (i) **Statistical feature attacks** [17, 33, 49] modify statistical features (flow duration and average packet size). These methods typically suffer from limitations in handling various features and generalizing across different attack scenarios. (ii) **Sequence feature attacks** [22, 30, 39, 48, 50] alter temporal and spatial sequence features (e.g.,

**Table 4: The adversarial effectiveness of DL-based traffic detection on the extended datasets.**

| | | | | | | | | | | | | | | | | | | |
|---|---|---|---|---|---|---|---|---|---|---|---|---|---|---|---|---|---|---|
| | | | | *(a) CICAPT-IIoT2024* | | | | | | | | | *(b) APT2024* | | | | | |
| Feature Extractor | ML Classifier | Detection | | | Attack (**ASR**)−*higher is better* | | | | | ML Classifier | Detection | | | Attack (**ASR**)−*higher is better* | | | | |
| | | P | R | F1 | T-F | W-L | BAE | R-T | **AdvTG** | | P | R | F1 | T-F | W-L | BAE | R-T | **AdvTG** |
| **Text** | TextCNN | 0.99 | 0.99 | 0.99 | 1.35% | 2.45% | 1.69% | 0.69% | **46.87%** | TextCNN | 0.98 | 0.99 | 0.99 | 0.30% | 0.60% | 0.60% | 1.40% | **53.97%** |
| | DeepLog | 0.99 | 1 | 0.99 | 1.91% | 1.74% | 3.45% | 5.15% | **33.59%** | DeepLog | 0.98 | 0.91 | 0.94 | 1.07% | 1.72% | 2.79% | 7.63% | **39.70%** |
| | BERT | 0.99 | 0.99 | 0.99 | 0.55% | 0.14% | 1.56% | 4.58% | **43.75%** | BERT | 0.99 | 0.99 | 0.99 | 1.11% | 1.61% | 2.32% | 4.64% | **50.78%** |
| **Image** | MalTraffic | 0.98 | 0.99 | 0.99 | - | - | - | 17.64% | **45.31%** | MalTraffic | 0.95 | 0.90 | 0.92 | - | - | - | 16.52% | **54.31%** |
| | RBRN | 0.98 | 0.99 | 0.99 | - | - | - | 18.79% | **43.75%** | RBRN | 0.97 | 0.97 | 0.97 | - | - | - | 20.68% | **52.73%** |
| | DeepMal | 0.98 | 0.99 | 0.98 | - | - | - | 15.08% | **42.18%** | DeepMal | 0.95 | 0.94 | 0.95 | - | - | - | 14.37% | **50.52%** |

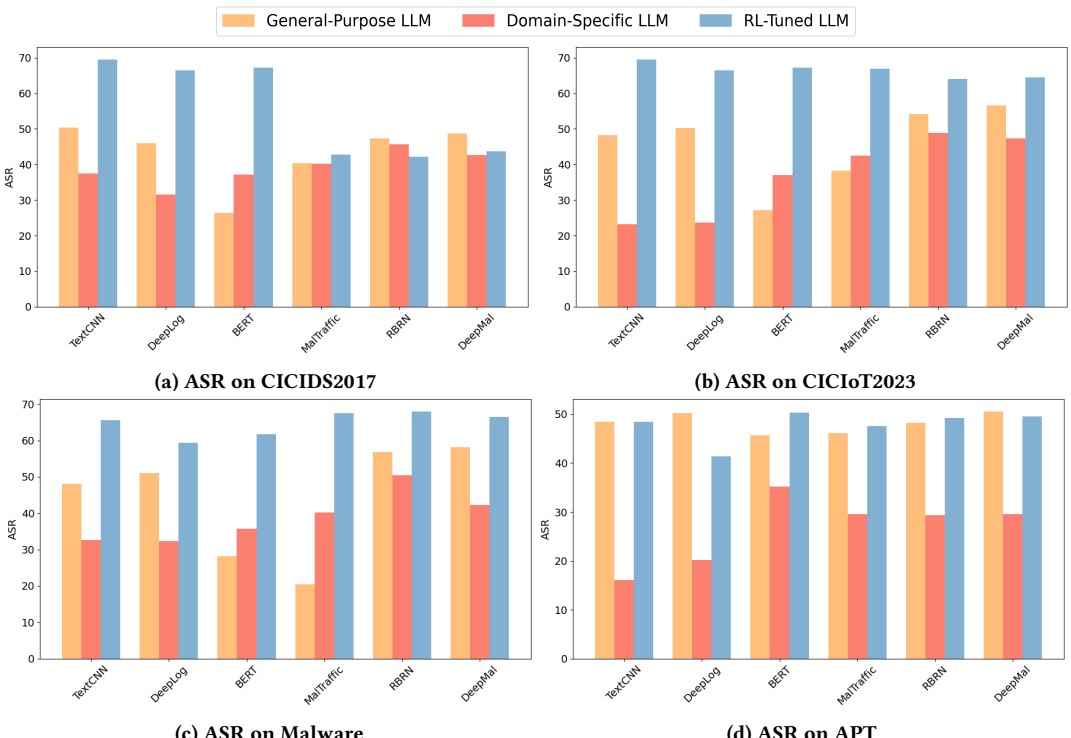

(a) ASR on CICIDS2017  (b) ASR on CICIoT2023

(c) ASR on Malware  (d) ASR on APT

**Figure 4: Adversarial effectiveness across different LLMs on various datasets and detection models.**

**Table 5: Effectiveness of adversarial attacks against LLM-based detection models**

| Detection Model | Datasets | P | R | F1 | ASR |
|---|---|---|---|---|---|
| ChatGPT | Malware | 0.71 | 0.54 | 0.61 | 16.35% |
| | APT | 0.89 | 0.42 | 0.57 | 16.64% |
| ChatGPTs | Malware | 0.79 | 0.56 | 0.66 | 16.03% |
| | APT | 0.90 | 0.43 | 0.58 | 15.57% |

packet intervals, packet lengths) but payload-based detection models remain effective. While they offer improvements in maintaining protocol compliance, they cannot often generate traffic content. (iii) **Content attacks** [10, 35, 51, 58] directly modify traffic packets to evade detection. These methods focus on generating traffic packets directly, but they struggle to produce traffic that is semantic, compliant, and functional.

## 10 CONCLUSION

In this paper, we propose AdvTG, a framework that uses LLMs and RL to generate adversarial traffic and deceive DL-based detection models that analyze malicious payloads. The adversarial traffic generated by AdvTG maintains both semantic coherence and protocol compliance. We introduce perturbations in the non-functional space to ensure the core functionality of the mutated traffic remains intact. Extensive experiments across various model architectures and feature sets demonstrate that AdvTG achieves a high ASR against DL-based models, highlighting the inherent weakness in these detection models.

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

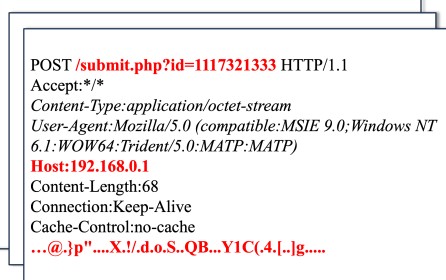

**Figure 5: Cobalt Strike HTTP traffic.**

# APPENDIX

## A MALICIOUS HTTP PACKET CASE

We use an example of Cobalt Strike's HTTP traffic to illustrate the distinction between functional and non-functional fields, as shown in Fig. 5. Cobalt Strike is a penetration testing tool often repurposed by attackers for malicious activities. The red-highlighted parts represent the core functional fields of the HTTP traffic, including the URI header, Host field, and payload. These are essential for maintaining traffic functionality and are difficult to modify without risking the disruption of network activity. While non-functional fields carry less significance compared to functional fields, they often contain inherent characteristics of traffic. As a result, non-functional fields are also frequently used as features to detect malicious traffic rather than solely relying on functional fields. The italicized parts show the *User-Agent* field (Mozilla/5.0 (compatible:MSIE 9.0;Windows NT 6.1:WOW64:Trident/5.0:MATP:MATP)) commonly used by Cobalt Strike and the *Content-Type* field (application/octet-stream), which is fixed features of Cobalt Strike. These are examples of non-functional features exploited by defenders for detection. Defenders can detect these features using rule-based methods or detection models. Similarly, attackers can modify non-functional fields to evade detection, especially DL-based detection.

## B A CASE OF DECEIVING DL-BASED DETECTION USING AdvTG

This example demonstrates the effectiveness of AdvTG in deceiving a DL-based malicious traffic detection. Initially, as shown in Fig. 6, the traffic packet on the left is classified with 97.85% confidence as malicious by the detection model. The packet includes a *User-Agent* field (Go-http-client/1.1), which is a known indicator of potentially malicious activity often associated with automated tools or attacks.

Through the application of AdvTG, modifications are made to certain non-functional fields of the traffic, ensuring that the traffic remains its original structure and functionality. Unlike traditional adversarial attack methods that often rely on manual feature manipulation, AdvTG leverages the ability of the LLM and RL to automate the generation of the most effective adversarial fields. By continuously refining the traffic through RL, AdvTG identifies and alters the non-functional fields that are most likely to deceive the detection model.

In the altered traffic packet (right side of the figure), the *User-Agent* field has been replaced with a more commonly used and benign browser identifier: Mozilla/5.0 (Windows NT 10.0; Win64; x64) AppleWebKit/537.36 (KHTML, like Gecko) Chrome/80.0.3987.149 Safari/537.36. This modification makes the traffic appear to be from a legitimate web browser, thus reducing the likelihood of detection. After this adversarial transformation, the detection model's classification confidence shifted dramatically. The modified packet is classified as 66.61% benign, a stark contrast to the initial malicious classification. This reduction in malicious probability highlights how adversarial attacks can successfully deceive well-trained DL-based models by subtly altering traffic fields that are commonly relied upon for detection.

**Table 6: Traffic datasets.**

| Dataset Group | Datasets | Attacks Types (Examples) | # Test Pkts (Malicious) | # Training Pkts |
|---|---|---|---|---|
| Base Dataset | CICIDS2017 CICIoT2023 Malware APT | Botnet, Web Attack, Infiltration Recon, Web attack, DDoS Trickbot, Qakbot, Emotet APT27, APT28, APT37 | 10,000 (5,132) 10,000 (3,045) 10,000 (2,670) 10,000 (4,813) | 100,000 (50,000 benign & 50,000 malicious) |
| Extended Dataset | CIC-APT-IIoT2024 APT2024 | APT29, APT28, APT32 Lazarus, Kimsuky, Bitter | 5,000 (2,500) 1,000 (384) | - |

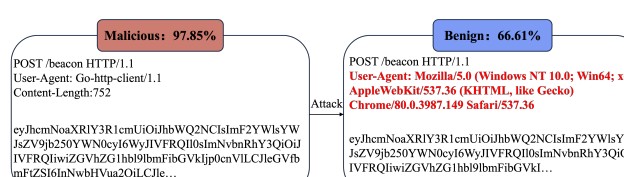

**Figure 6: A case of deceiving DL-based detection using Ad-vTG.**

## C  DETAILS OF DATASET

Table 6 summarizes the information about the traffic datasets used in this study, including the base dataset and extended database.

### C.1  Base Dataset

We utilize three publicly accessible, open-source datasets, along with an additional dataset that we collected independently as part of the base dataset.

- **CICIDS2017** [2] contains benign and the most up-to-date common attacks, which resembles real-world data. We extracted HTTP traffic from it, including Web Attack, Infiltration, Botnet, and others.
- **CICIoT2023** [3] is a novel and extensive IoT attack dataset to foster the development of security analytics applications in real IoT operations. The dataset contains a number of HTTP-based malicious attacks, including Recon, Web attack, DDoS, and others.
- **Malware** [4] is sourced from a prominent website that specializes in malware, containing traffic captured from popular malware between 2021 and 2023. HTTP traffic packets are extracted from 30 different categories of malware, including Trickbot, Qakbot, Emotet, and others.
- **APT** is the traffic generated by real APT organizations independently collected by our research team. First, we extract malicious sample hashes from 5000 APT reports, which include data from 256 different APT groups. Then these hashes are used to obtain the original traffic generated by these sample hashes from open-source sandbox intelligence such as Virustotal [47] and Anyrun [3]. We get a total of 10 gigabytes of traffic and filter out the HTTP traffic. Since this dataset is

---

[2]https://www.unb.ca/cic/datasets/ids-2017.html
[3]https://www.unb.ca/cic/datasets/iotdataset-2023.html
[4]https://www.malware-traffic-analysis.net/index.html

**Algorithm 1:** HTTP Traffic Compliance

**Input:** HTTP traffic: http_traffic
**Output:** Validation result: is_valid, Errors: errors
1  errors ← [ ]
2  parts ← Split http_traffic by "\r\n\r\n"
3  headers_part ← parts[0]
4  body_part ← if exists, parts[1]
5  headers_lines ← Split headers_part by "\r\n"
6  request_line ← headers_lines[0]
7  **if** *Invalid request line format* **then**
8     errors.append("Invalid request line")
9  **foreach** header *in* headers_lines[1 :] **do**
10    **if** *Invalid header format* **then**
11       errors.append("Invalid header")
12 **if** *Missing separating line between headers and body* **then**
13    errors.append("Missing separator")
14 **if** errors *is not empty* **then**
15    **return** False, errors
16 **return** True, "Valid HTTP traffic"

taken from a real sandbox, it more accurately reflects real-world conditions than open-source datasets.

- **Benign** consists of benign traffic extracted from several of the datasets above, as well as traffic we decrypted from the 300 most popular websites listed in the Alexa Top rankings.

### C.2  Extended Dataset

The extended dataset is not involved in any model training or fine-tuning; it is solely used for testing purposes.

- **CICAPT-IIoT2024** [5] is designed to provide cybersecurity researchers focusing on APT detection tasks with a comprehensive dataset specifically collected from an APT campaign in an Industrial Internet of Things (IIoT) environment. This dataset comprises a total of 19 APT groups, including well-known organizations such as APT28, APT29, APT32, and others.
- **APT2024** is a dataset we gathered from open-source sandbox environments, representing APT traffic data from the year 2024. This dataset consists of traffic patterns from 8 APT groups, such as Lazarus, Kimsuky, Bitter, and several others. These APT campaigns are critical for researchers aiming to enhance detection and mitigation strategies against advanced threats.

# D  HTTP TRAFFIC COMPLIANCE

As shown in algorithm 1, we define the compliance for HTTP traffic based on the RFC document. In simple terms, HTTP traffic should consist of the following parts: 1. Request Line: Includes the method (e.g., GET, POST, HEAD), the request URI, and the HTTP version. 2. Headers: Key-value pairs providing additional information about the request. 3. Blank Line: Separates the headers from the body. 4.

Body (Payload): Contains the data being sent. Through the above methods, we verify whether the generated traffic complies with the specifications. In addition, the generated HTTP traffic should remain functional. So we define something like the red part in Fig. 5 as unchangeable, and we force that part not to be changed.

---

[5]https://www.unb.ca/cic/datasets/iiot-dataset-2024.html