# OpenReview forum: "AdvTG: An Adversarial Traffic Generation Framework to Deceive DL-Based Malicious Traffic Detection Models"
_ACM.org/TheWebConf/2025/Conference — WWW 2025 Poster_

### Official Review · Reviewer_tQx9 · 2024-11-25

**Novelty:** 4
**Technical Quality:** 4

**Review:**

This paper proposes an adversarial traffic generation framework called AdvTG, based on LLM (Large Language Model) and RL (Reinforcement Learning), to deceive DL-based detection models. AdvTG takes functional fields as input and uses LLM to generate non-functional fields, forming complete traffic packets. The paper points out that existing adversarial attack methods for DL-based traffic detection can be categorized into three types: statistical feature attacks, sequence feature attacks, and content attacks. This paper focuses solely on content attacks, particularly deep learning detection models based on payload analysis.

**Pros:**

1. The overall logic of the paper is clear and easy to understand.
2. The main innovation of this work lies in the use of LLM and RL. The paper proposes generating adversarial traffic by fine-tuning LLM and applying RL to optimize the process of adversarial traffic generation.

**Cons:**

1. There seems to be a lack of critical experimental comparisons. As mentioned earlier, the paper categorizes existing adversarial traffic attack methods into three types and notes that it focuses only on content attacks. However, in the experimental evaluation section, I did not see a comparison with existing content attack methods targeting traffic.
2. In Section 7.1, the authors introduced four comparative attack baselines: TextFooler (T-F) [27], WordLevel (W-L) [61], BAE [21], and R-S. It is apparent that these methods are adversarial attack approaches in the text classification field, rather than for adversarial attacks targeting malicious traffic detection. Conducting experimental comparisons with other adversarial traffic attack methods, particularly those designed for malicious traffic detection, would better support the claims of this paper.

    [27]: 2020-AAAI - "Is BERT really robust? A strong baseline for natural language attack on text classification and entailment"

    [61]: 2020-ACL - "Word-level Textual Adversarial Attacking as Combinatorial Optimization"

    [21]: 2020-EMNLP - "BAE: BERT-based Adversarial Examples for Text Classification"


**Minor Errors:**

1. In Section 1 (Introduction), there is a repetition of the word "iteratively" in the contribution statement: *“enables the LLM to iteratively improve its outputs iteratively”*.
2. The title of Section 2.2, "THREAD MODEL," is incorrect; "Thread" should be "Threat.”
3. In Section 7.1 (Experimental Setup), the attack baseline *Random Substitution* is abbreviated as R-S, but in the subsequent experimental descriptions and tables, it is mistakenly referred to as R-T.
4. Some sentences are unclear or awkward. For example, in Section 7.2, the sentence *“We train models based on text and image features based on training sets”* should be revised for clarity.

**Questions:**

Please see the cons. Additionally, which specific fields are classified as functional fields, and which are categorized as non-functional fields in the paper?

**Reviewer Confidence:**

3: The reviewer is confident but not certain that the evaluation is correct

**Scope:**

3: The work is somewhat relevant to the Web and to the track, and is of narrow interest to a sub-community

---

### Official Review · Reviewer_NKXf · 2024-11-27

**Novelty:** 4
**Technical Quality:** 4

**Review:**

# Summary
This paper presents AdvTG, a novel adversarial traffic generation framework designed to deceive deep learning-based malicious traffic detection models. It addresses the limitations of existing adversarial attacks that tries to modify the content of packets, in particular payloads, while mantaining their semantics. AdvTG integrates large language models (LLMs) fine-tuned for traffic-specific tasks with reinforcement learning (RL) to optimize the generation of adversarial traffic samples.
The framework operates in three stages: training multiple Deep Learning (DL)-based detection models as attack targets, fine-tuning an LLM using traffic-specific data, and employing RL to refine adversarial traffic generation based on feedback from the detection models.
The experiment is comprehensive. It considers six datasets and six DL-based detection model. The results demonstrate that AdvTG is able to generate malicious traffic considerably more stealthy than other state-of-the-art malicious traffic techniques.

# Comments for authors
Thank you for submitting this paper! I think that the paper fits the interests of the community of this conference, given that it deals with malicious HTTP traffic generation and detection. The idea of the paper is good, even though the novelty is not particularly high, in my opinion. There are issues about the content and presentation that the authors should discuss and consider to address to make paper publishable.

## Strengths
The authors have done a good job highlighting the limitations of the current state-of-the-art malicious traffic generation methods and positioning their work in the literature (see Section 1) since modifying payloads of packets while maintaining the semantics is intuitively difficult. Moreover, I greatly appreciate the explanation of the motivations behind this work in Section 2. This work presents a certain grade of novelty, even though it is not excellent, since using LLMs to generate adversarial examples has already been explored in the literature [a]. In addition, fine-tuning a model to evade another is very reminiscent of generating a dataset for a surrogate model on black-box settings to mimic the (black-box) attacked model [b].
The proposal combines existing ideas adapted to the malicious traffic generation task instead of being a new fundamental contribution. However, using LLMs to generate adversarial traffic specifically has never been explored in the literature, at the best of my knowledge, so the idea is interesting. The considered threat model of the attack is reasonable since an attacker may not know details about the model used to perform the detection. Finally, the current experimental results are positive and show that fine-tuning the LLM to attack a specific detector significantly enhances the efficacy of the attack with respect to the considered baselines.

## Weaknesses

I appreciate the novelty of the idea, even though it is not excellent. However, there are some important issues about the technical quality of the paper to take into consideration.

First, I have a concern about the requirement to fine-tune the LLM with RL to make the generation of malicious traffic effective (as highlighted in the ablation study in Figure 4, Section 7.2). It is not clear how many queries are required for the target detector to perform the fine-tuning of the LLM. Making requests to a black-box model may have some financial costs for the attacker, as well as the attacker may want to keep the stealthiness of their attack, as highlighted by previous contributions in the adversarial machine-learning literature that try to minimize the number of queries to the attacked model to perform the attack [c]. However, the fine-tuning through RL that you propose requires querying the black-box detectors to gain a reward. You should also evaluate the number of queries used to perform an effective RL-based fine-tuning to be transparent about this point since it influences the feasibility of the proposed approach.

Second, in connection with the first issue, the authors should also discuss the transferability of their approach. Is the fine-tuned LLM for a specific detector able to generate traffic that also evades other detectors of the same type (image-based or text-based)? This is an important point to discuss and experimentally validate, in my opinion, since fine-tuning the LLM through RL can be computationally expensive. Moreover, the attacker know nothing about the attacked model, so they can suppose that the traffic detection model remains the same (e.g. when attacking several web applications of the same company).

Moreover, in Section 5, the authors declare to use parameter-efficient fine-tuning to improve the efficiency of the fine-tuning. However, its impact on the entire pipeline is not evaluated since no experiments about its efficiency are included in the paper. Without a validation of the importance of this building block of the framework, it is impossible to assess if it is relevant.

In the related work, the authors cite several other detectors of malicious traffic that have been not tested in the experimental evaluation. The reason behind excluding these baselines or why these are less representative of the chosen ones is not stated in the text. The authors should consider clarifying this point to make clear that the experimental methodology is fair.

Finally, the presentation is not satisfactory in its current state. There are some issues that the authors should consider to address when preparing the next version of their paper:
- Details about how data shown in Figure 1 are generated are missing, making it impossible to reproduce the experiment on the basis of them.
- It would also be informative to provide specific examples of prompts that you use to fine-tune the LLM used and an example of a query posed to the LLM to generate malicious traffic, as done in [a]. At the moment, the paper does not explore how these prompts are built, but it is one of the main points of your approach.
- The parameters used to train the defensive models are missing. It is important to specify them and how you have chosen them to ensure that you have followed the best practices from the corresponding papers.
- The authors should consider inserting a specific section that establishes the notation used. Some symbols are not contextualized appropriately. For example, the reward $r_\theta$ contains $\theta$ that is not defined (Section 6), a model is referred with both $h$ and $D_i$, a label is referred using both $y$ and $L_x$ and $\pi$ is not defined and written with undefined pedices and apices.

[a] Wang. et. al., Generating Valid and Natural Adversarial Examples with Large Language Models, in CSCWD 2024.
[b] Papernot et. al., Transferability in Machine Learning: from Phenomena to Black-Box Attacks using Adversarial Samples, in Arxiv 2016.
[c] Ilyas et. al., Black-Box Attacks with Limited Queries and Information, in ICML 2018.

**Questions:**

- What are the efficiency and number of queries required to fine-tune the LLM using RL?
- What about the transferability of the traffic generated by a LLM fine-tuned for a specific detector?
- What is the impact of Parameter-Efficient-Fine-Tuning in your proposed attack?
- Why have you not considered other malicious traffic detection baselines discussed in the related work in your experimental evaluation?
- See the issues about the presentation.

See the Weaknesses section of the paper for more details.

**Ethics Review Description:**

.

**Reviewer Confidence:**

3: The reviewer is confident but not certain that the evaluation is correct

**Scope:**

4: The work is relevant to the Web and to the track, and is of broad interest to the community

---

### Official Review · Reviewer_7RQF · 2024-11-30

**Novelty:** 6
**Technical Quality:** 6

**Review:**

This paper proposes AdvTG, a framework that uses LLMs and RL to generate adversarial traffic and deceive DL-based detection
models that analyze malicious payloads. The adversarial traffic generated by AdvTG maintains both semantic coherence
and protocol compliance. Authors introduce perturbations in the nonfunctional space to ensure the core functionality of the mutated
traffic remains intact. Extensive experiments across various model architectures and feature sets demonstrate that AdvTG achieves
a high ASR against DL-based models, highlighting the inherent weakness in these detection models.

The key challenges can be summarized into three categories.
• Generality. Traffic features should be mapped to real-world
traffic spaces rather than generating non-existent features.
• Availability. The generated traffic must adhere to strict protocol
compliance while maintaining its functionality.
• Payload Generation.

Four contributions such as
• AdvTG1, an adversarial traffic generation framework based on the LLM and RL to deceive DL-based malicious traffic models.

The paper is well organised with examples and experimental work. Three scenarios by answering the following key questions such as:
A. Why are we targeting content adversarial attacks?

**Questions:**

1. Detailed explanations of equations (1) and (2) in Image-based Features section are required.

**Reviewer Confidence:**

3: The reviewer is confident but not certain that the evaluation is correct

**Scope:**

4: The work is relevant to the Web and to the track, and is of broad interest to the community

---

### Official Review · Reviewer_F6GP · 2024-12-02

**Novelty:** 4
**Technical Quality:** 5

**Review:**

The authors present a method leveraging LLMs and reinforcement learning to generate adversarial traffic designed to evade deep learning-based detection models that analyze malicious payloads.

The overall concept is interesting and supported by various experiments conducted across multiple datasets. However, it is worth noting that the Targeted Detection Models and Attack Baselines used in the study are not derived from the most recent literature.

In conclusion, while the approach holds promise, incorporating more contemporary baselines and detection models could enhance the relevance and impact of the findings.

**Questions:**

1. The authors mention fine-tuning LLMs with domain-specific datasets. However, it would be helpful to provide details of the experimental setup for this fine-tuning process and also mention explicitly which datasets were used for fine-tuning. While the theoretical aspects of LoRA are discussed, key specifics such as the target modules, alpha, rank, and other hyperparameters are not mentioned in the setup section of the paper—unless I may have overlooked it.

2. I would like to seek clarification regarding the RL-tuned LLMs: are they general-purpose LLMs, or are they first fine-tuned using the approach mentioned in Section 5 and subsequently used explicitly for RL? Alternatively, is there another interpretation? I found it challenging to understand the full picture of the RL-based LLM and how equations 6 and 7 are applied in real-world tuning. This may be due to an unclear explanation in the writing and missing details in the setup outlined in Section 7.1. It would be beneficial if the authors could provide clarity on how the RL-tuning of LLMs was performed and explain the primary motivation behind it. Specifically, why is a domain-specific fine-tuned LLM that generates the packets not directly rewarded or penalized using RL?

3. I would suggest authors write some considerations as the concepts in the framework aim to generate adversarial examples using LLMs.

**Reviewer Confidence:**

3: The reviewer is confident but not certain that the evaluation is correct

**Scope:**

4: The work is relevant to the Web and to the track, and is of broad interest to the community